# Feeding the Brain: Effect of Nutrients on Cognition, Synaptic Function, and AMPA Receptors

**DOI:** 10.3390/nu14194137

**Published:** 2022-10-05

**Authors:** Rut Fadó, Anna Molins, Rocío Rojas, Núria Casals

**Affiliations:** 1Basic Sciences Department, Faculty of Medicine and Health Sciences, Universitat Internacional de Catalunya, E-08195 Sant Cugat del Vallès, Spain; 2Centro de Investigación Biomédica en Red de Fisiopatología de la Obesidad y la Nutrición (CIBEROBN), Instituto de Salud Carlos III, E-28029 Madrid, Spain; 3Institut de Neurociències, Universitat Autònoma de Barcelona, Bellaterra, E-08193 Cerdanyola del Vallès, Spain

**Keywords:** diet, cognition, memory, learning, hippocampus, synaptic function, neuroplasticity, BDNF, AMPARs, neurodegeneration

## Abstract

In recent decades, traditional eating habits have been replaced by a more globalized diet, rich in saturated fatty acids and simple sugars. Extensive evidence shows that these dietary factors contribute to cognitive health impairment as well as increase the incidence of metabolic diseases such as obesity and diabetes. However, how these nutrients modulate synaptic function and neuroplasticity is poorly understood. We review the Western, ketogenic, and paleolithic diets for their effects on cognition and correlations with synaptic changes, focusing mainly (but not exclusively) on animal model studies aimed at tracing molecular alterations that may contribute to impaired human cognition. We observe that memory and learning deficits mediated by high-fat/high-sugar diets, even over short exposure times, are associated with reduced arborization, widened synaptic cleft, narrowed post-synaptic zone, and decreased activity-dependent synaptic plasticity in the hippocampus, and also observe that these alterations correlate with deregulation of the AMPA-type glutamate ionotropic receptors (AMPARs) that are crucial to neuroplasticity. Furthermore, we explored which diet-mediated mechanisms modulate synaptic AMPARs and whether certain supplements or nutritional interventions could reverse deleterious effects, contributing to improved learning and memory in older people and patients with Alzheimer’s disease.

## 1. Introduction

The growing prevalence of cognitive impairment and dementia, with 100 million cases estimated by 2050, is becoming a serious public health concern with both individual and socioeconomic burdens [1]. Although the rise in life expectancy is one of the main causes of the increased prevalence, multiple other factors are likely to play a role. In recent years, increasing attention has been paid to the impact of diet on brain health and mental function [2,3], with several dietary components, such as carbohydrates, fats, and hormones, reported to influence cognition [1]. Indeed, the consumption frequency of healthy or unhealthy foods perfectly correlates with better or worse cognitive performance in older adults [4]. It would seem, therefore, that our diet directly affects brain health and the probability of developing dementia and neurodegenerative disorders in later life.

The consumption of high-fat/high-sugar diets in modern civilizations, combined with overeating behaviors and sedentary lifestyles, has led to exponential growth in obesity cases worldwide, reported to be 650 million people in 2016, according to the World Health Organization (WHO) [5]. Obesity increases the risk of developing dementia in later life [6,7], and has been associated with immediately altered cognitive performance, including impaired verbal learning, working memory, stimulus reward learning, and executive functions (adaptation to novel situations, cognitive flexibility, attention span, planning, and judgment) [8,9,10]. Obesity is also considered a risk factor for developing multiple comorbidities, such as type 2 diabetes (T2D), hypercholesterolemia, hypertension, and metabolic syndrome (MetS), all of which may have an independent impact on cognition. In fact, T2D (with 422 million cases in 2014, according to the WHO, [11]) correlates with cognitive dysfunction, including reduced information processing speed and altered verbal, visual memory, and executive functions [12,13,14]. Even impaired glucose tolerance, which develops prior to T2D, is considered a risk factor for cognitive damage [15]. In addition, also associated with a higher risk of cognitive impairment and dementia is MetS [16,17,18], defined as a multifactorial disorder characterized by abdominal obesity, insulin resistance, hypertension, and hyperlipidemia [19]. However, still poorly understood are the physiopathological mechanisms by which metabolic diseases affect learning and memory processes, and raise the risk of late-life dementia and neurodegeneration.

During learning and memory task performance, neural networks, in a process called neuroplasticity [20], make new connections and/or adjustments in synaptic strength, especially in the hippocampal glutamatergic circuits. This reorganization is carried out by tight control of the number of α-amino-3-hydroxy-5-methyl-4-isoxazolepropionic acid (AMPA)-type glutamate ionotropic receptors (AMPARs) in the post-synaptic zone (PSZ). Since glutamate is the major excitatory neurotransmitter in the central nervous system, AMPARs are the main mediators of fast synaptic transmission. Functional receptors are tetrameric assemblies of subunits GluA1-4. Whereas GluA4 is mostly expressed during early development stages, in mature excitatory neurons, AMPARs contain two dimers of GluA1/GluA2 (~80% of all synaptic AMPARs) or GluA2/GluA3 heteromers [21]. In resting conditions, the presence of GluA2 in surface AMPARs makes these Ca^2+^-impermeable [20]. Following stimulation with glutamate, precise regulation leads to synaptic incorporation of GluA1-GluA1 homomers. These GluA2-lacking AMPARs are Ca^2+^-permeable (CP-AMPARs), and, over a restricted period of time, induce Ca^2+^-sensitive signaling events that sustain synaptic potentiation or modulate subsequent neuroplasticity. However, aberrant incorporation of CP-AMPARs can result in excessive Ca^2+^ influx, inducing excitotoxicity and cell death, as reported for brain diseases [22]. An excitotoxic influx of Ca^2+^ is also mediated by overactivation of N-methyl-D-aspartate (NMDA)-type glutamate ionotropic receptors (NMDARs) [23]. Functional NMDARs are tetramers of GluN1 (A or B) and GluN2 (A-D), and are also involved in neuroplasticity modulating synaptic incorporation of CP-AMPARs.

Changes in the expression of a particular subunit of AMPARs produce an imbalance in AMPAR tetrameric composition or dysregulate its trafficking toward the PSZ, and both have been linked to Alzheimer’s disease (AD) and cognitive aging [21,24]. In AD, high levels of amyloid-β oligomers (Aβ)—a main pathological hallmark, together with Tau phosphorylation—induce readjustments in the synaptic insertion of CP-AMPARs [25,26]. Soluble Aβ oligomers reduced surface expression of AMPARs and blocked their extrasynaptic delivery mediated by chemical synaptic potentiation in neuronal cultures [26]. However, intracellular infusion of oligomeric Aβ enhanced AMPAR-mediated synaptic transmission in hippocampal slices, but the knockdown of GluA1 prevented this effect, suggesting that intracellular Aβ induced synaptic insertion of CP-AMPARs [25]. In any case, Aβ dysregulates AMPAR trafficking [27] and AMPAR-mediated synaptic plasticity impairment is one of the early onsets of AD (reviewed by [21]). This early synaptic dysfunction largely precedes any sign of cognitive decline, in both neurodegenerated and aged brains [21,28]. The fact that impaired learning and memory processes in AD and aged people are exacerbated by unhealthy diets [29], underlines the impact of nutritional factors at the synaptic level. Additionally, dynamic regulation of AMPARs has been associated with different metabolic situations, orexigenic and anorexigenic hormones (like leptin and ghrelin), and dietary factors [30,31,32,33,34].

In summary, it is becoming evident that dietary habits and specific nutrients play an essential role in brain health. However, solid evidence of nutritional control of synaptic function focused on AMPARs and its correlation with cognitive performance is not completely understood.

Therefore, our main aim in this narrative review was to comprehensively compile, from the literature, analyses of the effects of the main dietary habits and nutritional factors on cognition, synaptic function, and AMPAR dynamics. In particular, in an endeavor to clarify conflicting findings, we review the effects on learning and memory processes of diets with high sugar, fat, and/or protein content (such as the Western, ketogenic, and paleolithic diets), focusing especially on nutrient composition, exposure time, and differences between sexes and age groups. Focusing especially on readjustments in synaptic AMPARs, we also document existing evidence on the association between early dysfunction in hippocampal synapses and cognitive impairment, summarizing the pathways that may explain nutrient-mediated modulation of neuroplasticity. Disentangling the molecular mechanisms that underpin the damaging or neuroprotective effects of certain nutrients may help design better dietary interventions and therapeutic approaches, both to the prevention of cognitive decline linked to aging and to neurodegenerative and metabolic diseases.

## 2. Western Diet

In recent decades, most societies have gradually abandoned dietary traditions to globalize their lifestyles. In southern Europe, for instance, the Western diet (WD), featured by high animal-based food, fat, and simple sugar intake, is gradually replacing the primarily plant-based Mediterranean diet (MedDiet), rich in vegetables, fruit, whole grains, olive oil, and fish, and low in red meats and refined sugars [35], and associated with a lower rate of cognitive function decline [36].

It has been documented that diets high in either fats or simple sugars have deleterious effects on cognition (discussed below), so the combination of both in the WD evidently contributes to memory and learning impairments. In lean young humans, a 4- or 8-day intervention with a Western style-based breakfast (high in saturated fats and added sugars) was enough to reduce hippocampal-dependent learning and memory [37,38]. In a longitudinal study of elderly people, the WD was a predictor of poorer memory and processing speed [39]. There is, however, little literature analyzing the effects of the WD and its components on brain function in humans [40], which is why we revisit experimental approaches with animal models.

### 2.1. Cognition Studies in Animal Models

While several high-fat diet (HFD)/high-sugar diet (HSD) protocols in animal models aim to mimic the WD in humans, many factors determine cognitive outcomes, such as the nature and percentages of fats and sugars (in WD-mimetics approaches, and in healthier control diets typically composed of 45–65% carbohydrates and 5–15% fats), palatability, exposure over time, individual susceptibility or resistance to obesity, and cognition tests used to measure learning and memory. With just a few exceptions, in which no differences were reported at 5, 10, or 12 weeks of the WD [41,42,43], a wide range of experimental models have demonstrated negative WD effects on learning and on short- and long-term memory, the details of which are summarized in Table 1.

As happens in humans, short exposition to Western style-based breakfast was enough to reduce memory in rodents. Five days of exposure to a non-standardized cafeteria diet (CAFD) composed of cakes, biscuits, and a protein source, plus a high-sucrose (HSu) solution, compromised hippocampal-dependent place recognition memory in adult rats, although other memory tasks remained unaffected [49]; comparable deficits were also reported for the CAFD without the HSu solution and the HSu solution without the CAFD. In young rat females, 5 weeks of CAFD or high-fructose solution (HFru) intake induced similar memory impairments, and intervention with a healthier diet was only able to revert HFru-mediated, but not CAFD-mediated deficits [44]. In adult mouse males, 14 weeks of combined HFD and HFru (HFD-HFru) intake resulted in a poorer cognitive performance than with an HFD alone [69]. To the best of our knowledge, there are no studies proving whether longer-term consumption of an HFD or HSD have additive effects in memory loss and synaptic function. However, we believe that this is likely the case, since high intake of both would simultaneously damage different signaling pathways with the same result.

Especially relevant is the fact that this brain damage is not equal in both sexes. In adult animals, females seem to be more resistant to WD-induced cognitive impairment than males [48,71]. Males are reported to consume a high percentage of calories from fats and proteins, while females show a greater preference for sweet-related calories [48]. However, sexual dysmorphism in nutrient choice is not the only reason to explain sex-specific consequences of the WD on cognitive outcomes, as sex hormones play an important role. In male rats, gonadectomy reverted HFD-HFru effects on some learning and memory tasks, but did not improve or even worsen the effects in females [71]. In non-human primates on an obesogenic diet, estrogen replacement therapy after an ovo-hysterectomy increased cognitive performance [73]. More in-depth studies analyzing sex- and hormone-dependent effects of an HFD or HSD on synaptic and cognitive injury could lead to better diet interventions for patients with MetS, especially for women after the menopause.

There is hope in the fact that WD-induced impairment in memory seems to be reversible, as demonstrated for male and female young animals switched from 4 months on an HFD/HSD to a more balanced control diet for 6 weeks [52,65], and also for male adult animals on a 3-week WD followed by a 3-week control diet [62]. By contrast, AD-related pathology would be exacerbated by the WD, which has been shown to accelerate age-associated cognitive decline in AD animal models (as demonstrated, for example, in [74,75]).

Moreover, the cognitive dysfunction associated to MetS can be countered by life-style strategies in clinical and preclinical studies, including healthy dietary habits, some dietary supplements (like vitamins and antioxidants), reduced calorie intake, and exercise [76,77], whereas no medications successfully treat cognitive impairment in these patients. Neither antihypertensive, statins nor drugs for AD have revealed efficiency on slowing cognitive deficits in preclinical models of MetS [78,79]. By contrast, high adherence to the MedDiet decreased the risk of developing cognitive impairment and dementia even in individuals with established MetS [77,80]. Accordingly, meals with a high content in fruits and vegetables, or in fish, exhibit a significant inverse association with cognitive decline in the general population and have a preventive role in cognition-related alteration associated with MetS [81,82]. In summary, healthy nutritional habits and lifestyle interventions are nowadays the best therapeutic strategy to reduce MetS components and prevent or delay cognitive decline.

### 2.2. Synaptic Function and Neuroplasticity

WD-compromised acquisition of new information in learning and memory processes perfectly correlates with deficits in hippocampal dendrites, synapsis morphology, and activity-dependent functional plasticity, as exemplified by the long-term potentiation (LTP) paradigm. LTP is an electrophysiologic high-frequency stimulation protocol that mimics physiologic activity during learning and memory [20]. This stimulation increases the number of CP-AMPARs (GluA1-GluA1 tetramers) in the PSZ, inducing trafficking to the cell surface via the recycling pathway and lateral movement from perisynaptic sites, and enhancing synaptic transmission efficiency. It has been reported that long-term WD-impaired synaptic plasticity in the hippocampus (HPC) reduces dendritic spine density and LTP [54,55,64], and also that the WD alters dendritic arborization, which is reduced in the HPC and increased in the entorhinal cortex [45]. This last finding could be a compensatory effect due to synaptic loss in distal regions like the HPC or even in proximal synapses, since the WD has also been reported to impair episodic memory in this particular case.

WD-mediated cognitive impairment is associated with deficits in certain synaptic proteins. Levels of the main neurotrophin, i.e., brain-derived neurotrophic factor (BDNF), were reduced in the HPC of adult mice after long-term exposure (2–8 months) to the WD [47,54,63,64,66,70], while no differences were observed for shorter exposure (20–42 days) [46,49,51]. A decrease in the BDNF receptor, tropomyosin receptor kinase B (TrkB), has also been reported [55,59]. Interestingly, not all nutritional sources alter BDNF levels in the same way. While a high-dextrose HFD downregulated BDNF in both the HPC and prefrontal cortex (PFC), this was not the case for a high-sucrose HFD [70].

Moreover, proteomic analysis of the dorsal HPC of rats on the WD for 8 weeks revealed alterations in synaptic plasticity markers, including a specific reduction in AMPAR subunit GluA3 [53]. In fact, selectively decreased synaptic GluA3 has also been shown to be involved in dendritic spine loss in frontotemporal dementia [83]. Other long-term WD paradigms demonstrate reductions in other glutamate receptors, including the AMPAR subunit GluA1 (at 13 weeks) [59] and the NMDAR subunit GluN2A (at 10 weeks) [42]. The decrease in GluA1 may occur because BDNF regulates its synthesis [84]. At synaptic sites, GluN2A-lacking NMDARs may display altered synaptic plasticity, as described for its deletion in pyramidal cells from HPC slices [85]. Lower levels of glutamate and its precursor, glutamine, have also been detected in the HPC of HFD-HFru- and HFD-fed mice, while gamma-aminobutyric acid (GABA), the most common inhibitory neurotransmitter, remained unaltered [69]. Short-term studies are needed, nonetheless, to understand whether the deleterious effects of the WD on memory loss are a cause or consequence of changes in the glutamate neurotransmitter and its receptors.

### 2.3. Other Mechanisms

Other mechanisms are also compromised in WD-induced cognitive damage: insulin (INS) signaling in the brain [60,68], blood-brain barrier (BBB) permeability [41,57,62], HPC inflammation [45,46,49,54,60,68], microglia activation [54,57,65,68,72], endoplasmic reticulum (ER) and oxidative stress [43,51,58,60,61,71], gut microbiome composition [44,46,54,56,58], and neurogenesis [59]. Therapeutic strategies based on modulating these mechanisms have demonstrated unequal effects on memory loss.

Insulin supplementation has been shown to prevent WD-reduced neuroplasticity [54], and treatment with metformin, the first line drug for T2D, improved cognitive function in HFD-HFru-fed mice, but failed to restore BBB permeability [57]. It remains unclear whether the loss of BBB permeability mediated by the WD is a cause of memory loss. It has been reported that an HFD-HFru produced BBB dysfunction after 4 weeks, but no significant decline in learning and memory [72]. In contrast, other studies, using different HFD/HSD paradigms, showed cognitive impairment after shorter exposure to the WD [41,49,63], in one case, reporting increased BBB permeability in WD-induced obese subjects at 12 weeks, but not in WD-resistant obese subjects at 5 weeks [41]; this would suggest that BBB leakage could, depending on exposure time, diet composition, and obesity phenotype, be a consequence of the WD.

Furthermore, pre-exposure to a probiotic called VSL#3 prevented WD-induced memory deficits for an HPC-dependent task, but caused deficits for a perirhinal-dependent task, irrespective of diet and dose [50], suggesting that this probiotic may be detrimental for healthy subjects. In one study, treatment with the antibiotic minocycline, which alters microbiome composition, prevented or reverted CAFD-induced spatial memory impairment [46]; however, another study showed that dietary intervention restored CAFD-dependent memory loss, but not gut microbiota alterations [44]. Finally, vitamin E administration normalized the effect of the CAFD on HPC oxidative stress markers and prevented diet-mediated memory decline [51]. The above are just some examples of efforts focused on discovering new drugs and/or bioactive foods with neuroprotective properties that could counteract HFD/HSD-induced brain damage.

In conclusion, an association between WD and alterations in memory and learning processes has been demonstrated both in human studies and experimental approaches on animal models. Such effects correlated with synaptic and neuroplasticity dysfunctions in the hippocampus, corresponding with reduced dendritic spine density and arborization, decreased LTP, and deficits in synaptic proteins, like AMPARs. Furthermore, other mechanisms regarding INS signaling and BBB permeability, among others, appeared to be affected too. Fortunately, WD-induced cognitive decline seems to be reversible, as shown in animal studies that switched from an HFD/HSD to a more balanced control diet. However, more in-depth investigations are required to fully understand the effect of WD and its components on brain function to create better diet interventions to ameliorate cognitive alterations.

Interestingly, more knowledge about the effects of nutrients at cognitive, synaptic and neuroplasticity level can be achieved reviewing current evidence with HFD alone (especially considering the fat type) and HSD alone (comparing fructose and sucrose intake), as documented in the following chapters.

## 3. High-Fat Diet

In humans, combined high-fat/high-sugar intake (as in the WD) is more frequent than an HFD alone (35–60% calories from fats), yet it is difficult to obtain a real picture of HFD impact on cognition, due mainly to variations between studies in terms of methodology, study type, target population, self-reported diet intake, measured neuropsychological tasks, and the limited number of articles included in quantitative analyses.

### 3.1. Cognition Studies with Mixed and Saturated Fats

While it has been widely reported that the WD impairs memory in the general population, an HFD alone failed to reach statistical significance in the latest systematic review [40]. Although cognitive performance is affected differently by the fat type (animal- or plant-based), not all the studies specified type, making it more difficult to draw reliable conclusions. Although excessive fat intake can have detrimental effects on cognition at any age, certain life stages seem to be especially sensitive, namely, the adolescent and elderly stages; higher habitual intake of unhealthy saturated fats (solid fats that come mainly from animals, such as meat and dairy) and trans fats (liquid oils that become solid in processed foods) was more clearly associated with impaired memory performance in young and older age groups than in healthy adult groups [86,87,88,89,90]. Moreover, contributing more to cognitive decline than excessive calorie intake is exposure to certain types of obesogenic dietary components, like saturated and trans fatty acids [91]. More controlled experimental approaches with HFD-fed animals models would throw light on the precise effect of fats on synaptic function, as well as the cellular and molecular mechanisms involved in cognitive decline.

As indicated by the animal studies summarized in Table 2, both young and aged animals are, like humans, susceptible to HFD-induced brain damage. Although the nature of the fats used is not always reported, in most cases, an HFD typically consists of a mix of fats, with a predominance of saturated fats over unsaturated fats. Apparently, 3-days consumption of a mixed HFD by aged animals was enough to damage HPC-dependent long-term but not short-term memory, while younger adults remained unaffected [92]; however, another study has demonstrated that short-term memory was also impaired after 7 days on a mixed HFD [93]. In neurodegeneration linked to AD, a mixed HFD induced long-term memory impairment, but no changes in short-term memory [94]. Moreover, early life exposure to a mixed HFD produced learning and memory deficiencies in aging females, even if a healthier diet was consumed during adulthood [95]. Thus, both early and late life stages are particularly vulnerable to obesogenic diets, which initially affect long-term, then short-term memory. However, it is important to note that a long-term mixed HFD impaired cognition even in adulthood [96,97,98,99]. Furthermore, foods enriched with saturated fat, trans fat, and/or cholesterol, even over short periods, had deleterious effects on the memory of young, adult, and aged animals when compared with iso-caloric control diets [100,101,102,103].

### 3.2. Cognition Studies with Polyunsaturated Fats

It is widely accepted that dietary unsaturated fats are healthy in contraposition to saturated fats. For instance, memory in adult mice was impaired after exposure to a saturated HFD, but not to an isocaloric diet with polyunsaturated fatty acids (PUFAs) [100]. However, it is important to differentiate between the unsaturated fats since their effects on the body differ. The most abundant omega-3 (ω-3) and omega-6 (ω-6) PUFAs in the brain are docosahexaenoic acid (DHA; 22:6) and araquidonic acid (AA; 20:4), respectively, mostly obtained from diet or synthesized in the liver from dietary alpha-linolenic acid (ALA, 18:3) and linoleic acid (LA, 18:2), respectively. The most abundant monounsaturated fatty acid (MUFA), oleic acid (OA, 18:1), is found in food and in the mammalian brain [133].

Several studies have been conducted to clarify the role of the different unsaturated fats in human cognition. A systematic review showed that diet supplementation with ω-3 PUFAs is associated with cognitive improvement in patients with AD, but only at early stages of the disease when the impairment is very mild [134], and it is not clear whether ω-3 or ω-6 PUFAs can prevent cognitive decline in healthy elderly people [135]. Moreover, ω-3 supplementation during pregnancy, breastfeeding, and the early years of life did not seem to improve cognitive skills in children. Additionally, diet supplementation with DHA did not seem to improve working memory performance in healthy young adults [136]. However, combined supplementation with ω-3, carotenoid and vitamin E or with long-chain ω-3 fatty acids alone exerted positive effects on working memory tasks in cognitively healthy older adults [137,138] and in patients with MetS (and with schizophrenia) [139]. Furthermore, virgin olive oil, which is rich in OA, shows anti-aging and neuroprotective effects [80], although this seems to be due, not to OA *per se*, but to other minor components present in extra-virgin olive oil, such as polyphenols (with antioxidant activity) [140]. Interestingly, adherence to the MedDiet (especially rich in both OA and ω-3) reduces the risk of mild cognitive impairment, progression to dementia, and even the AD mortality rate (for a review, see [141]). The MedDiet, including equivalent nutritional interventions with different food bases carried out in non-Mediterranean countries (e.g., Japan or Australia), has been associated with improved learning and memory parameters during aging.

Since studies in humans are complex in terms of implementation and of consistent results with specific nutrients, animal studies are key to addressing nutrition questions. In older mice, the daily administration of an ω-3 mix, but not olive oil, for 2 months improved cognitive performance [142], while perilla oil (rich in ω-3) more than corn oil (rich in ω-6) improved spatial learning and memory in rats [143]. A study to test 14-day plant oil supplementation showed attenuated deleterious effects in different animals previously brain-injected with Aβ [144]. Interestingly, only perilla oil slightly improved cognitive performance, while ω-6 supplementation seemed to potentiate the harmful effects of Aβ42 [145]. Olive oil supplementation for 6 months improved cognitive performance in a tauopathy animal model [146], although, since virgin olive oil contains different micronutrients, it cannot be affirmed that the beneficial effects were exclusively due to OA, and no studies exist specifically of OA.

### 3.3. Synaptic Function and Neuroplasticity with Mixed and Saturated Fats

Strong evidence exists for HFD detrimental effects on synaptic function. Mixed or saturated HFD (7 days or 46 weeks) reduced branching and spine density in the HPC and PFC in both adolescent and aged animals [101,124]. In adult mice, lesser post synaptic density and a wider synaptic cleft were reported for both acute and long-lasting HFD exposure [93,99]. In line with in vivo findings, palmitic acid (PA; 200 µM) reduced dendritic length and arborization in cultured HPC neurons, but the effects of PA were counteracted when combined with the ω-3 DHA (125 µM) [107]. Similarly, DHA (200 µM) co-exposure prevented PA-(200 µM) mediated synaptic loss and dendritic blebbing [147]. Taken together, those results suggest that the intake of unsaturated and saturated fatty acids (in a proportion close to 1:1) could reverse deleterious cognitive effects, and may explain why mixed HFD sometimes do not impair memory [122].

Moreover, numerous studies have demonstrated that both PA treatment and a mixed or saturated HFD inhibit LTP in HPC slices [96,102,103,104,117,124], whereas no effect has been observed in the dentate gyrus or in long-term depression (LTD) [104,120]. LTD, as the opposing process to LTP, is an activity-dependent reduction in synaptic strength that underlies surface elimination of AMPARs [20]. In another study, saturated oil-enriched food potentiated LTD and inhibited LTP more strongly than unsaturated oil-enriched food [32], suggesting that only saturated fats are detrimental. This enhancement in LTD may be associated with spine shrinkage and AMPAR synaptic loss.

In vitro, treatment with PA reduced AMPA sensitivity in cortical neurons, while no changes were observed for NMDA [147], suggesting that PA may specifically reduce AMPAR synaptic levels. It was also demonstrated that PA-dependent morphological changes (cell body size) in HPC neurons are mediated by phosphoinositide 3-kinase (PI3K) [147]. The fact that this kinase phosphorylates the plasma membrane (PM) phosphoinositide PI(4,5)P_2_, a lipid that directly binds and stabilizes surface GluA1 [148], suggests that PA may downregulate the synaptic levels of GluA1 subunit through that signaling pathway.

In vivo, early HFD exposure induced downregulation of the Ca^2+^-impermeable GluA2 subunit and its scaffold postsynaptic density protein 95 (PSD95) in the HPC [109]. Additionally, in young animals, HFD exposure (rich in saturated fats) increased PA deposition in the HPC, leading to hyper-palmitoylation and hypo-phosphorylation of GluA1, reducing its recruitment at the PM and inhibiting AMPAR-mediated currents [32]. In adult animals on a long-term mixed HFD, surface GluA1 levels in the HPC were increased, but no changes were observed in total levels [125]. As explained above, enrichment of synaptic CP-AMPARs (mainly GluA1 homomers) has been linked to pathological conditions by contributing to excessive Ca^2+^ influx, which ultimately causes synaptic dysfunction [20]. It is unknown whether this long-term increase in surface GluA1 may be a consequence of deregulation in scaffold proteins like PSD95, or in their trafficking towards the PM. Figure 1 depicts mixed and saturated HFD deregulation of AMPARs in the HPC.

The expression of other glutamate-type receptors (GluN1, 2B, and 2A) and of pre- and post-synaptic markers was frequently reduced by a mixed HFD, with PSD95 seeming to correlate better with impaired activity-dependent neuronal plasticity and memory deficits [93,97,98,99,102,109,126,128,130]. Like GluA1, the palmitoylation of other synaptic proteins plays an important role in regulating their membrane localization and AMPAR trafficking, and in controlling neuronal development, neurotransmission, and synaptic plasticity [149]. It would be interesting to test whether excessive palmitoylation of these proteins could underlie HFD-induced post-synaptic damage, as has been proved for neurodegenerative diseases.

Mixed and saturated HFD-induced impairment is also associated with changes in other neuroplasticity-related factors, such as decreased BDNF [93,95,97,106,113,115,123]. The fact that acute exposure to an HFD leads to changes in BDNF levels suggests that the alterations may be a direct cause of glutamatergic synaptic dysfunction. Additionally, and as explained above, it should be noted that this decrease is only detected for the WD after long-term exposure, suggesting that a greater amount of saturated fats correlates with a greater impact on BDNF levels. Moreover, therapeutic approaches that rescue BDNF signaling (physical exercise, calorie restriction, and intermittent fasting) have been associated with cognitive improvement (for reviews, see [150,151]).

### 3.4. Synaptic Function and Neuroplasticity with Polyunsaturated Fats

According to the observed effects of these fatty acids on cognition, a mix of ω-3 PUFAs supplementation in aged mice for 2 months increased neurogenesis and dendritic arborization of newly generated HPC neurons [142]. Moreover, ω-3 but not ω-6 supplementation increased dendritic spine density and the expression of different synaptic proteins in the HPC [143,152], while ω-3 supplementation in rat dams during pregnancy and lactation increased the dentate gyrus LTP of the pups [153], while ω-3 deficiency in pups for 5 months after weaning reduced LTP in the CA1-HPC [154].

Regarding the expression of glutamate receptors, effects driven by the different fatty acids correlate with cognitive and synaptic function outcomes. ω-3 or perilla oil supplementation, compared to ω-6-rich diets, increased AMPARs in the HPC and reversed aging-mediated depletion of AMPARs in different brain regions [143,145,152,155]. ω-3 supplementation also increased BDNF expression and reduced apoptosis and astrocytosis [142,144]. In contrast, olive oil did not change total AMPAR expression or GluA1 phosphorylation [145,146]. Moreover, in mice pups, ω-3 deficiency for 5 months after weaning reduced GluA1 and GluN2B in HPC synaptosomes [154]. Altogether, those studies suggest that ω-3 PUFAs improve synaptic function, while ω-6 PUFAs have no or even a negative effect, while the few studies on the effects of OA alone do not allow any conclusions to be drawn. Figure 2 depicts the role played by ω-3 in protecting against neurodegeneration and aging.

### 3.5. Other Mecanisms of Hippocampal Damage with Mixed and Saturated Fats

Altered brain insulin signaling is also an important hallmark in mixed or saturated HFD-mediated deficits [32,94,114,124,131,132]. PA treatment induced over-activation of INS signaling [147] and INS resistance [32] in HPC neurons. In contrast, metformin treatment attenuated INS resistance and restored learning behavior [120,124,127]. Similarly, an HFD caused leptin resistance, blocking leptin-induced potentiation of AMPAR-mediated synaptic transmission in HPC neurons [110]. Therefore, anti-diabetic treatments, which attenuate cognitive deficits in high-energy-fed rats, could be interesting prophylactic options for patients with MetS.

Mixed or saturated HFD-mediated synaptic dysfunction of the HPC has also been associated with other mechanisms: increased oxidative stress [94,98,117,121,132], mitochondrial dysfunction [124], ER stress [114,115,156], decreased glucose transport [132], reduced neurogenesis [111,132], metabolism and gut microbiota alterations [93,97,100], glucocorticoid axis defects [103,109], and microglia activation and neuroinflammation [92,93,97,98,99,101,105,107,108,111,117,127,129]. For instance, central administration of an IL-1 receptor antagonist or intraperitoneal administration of an anti-TNF-α antibody rescued mixed HFD-induced cognitive deficits [92,105], systemic glucocorticoid receptor antagonist administration alleviated LTP and memory impairments mediated by a saturated HFD [103]. However, it is still unclear how those mechanisms contribute to HFD-mediated synaptic dysfunction and cognitive decline.

Furthermore, the role of exosomes in neurodegenerative diseases is a very exciting emerging field, although the mechanisms underlying exosomes-mediated neuroplasticity and AMPAR regulation are still being uncovered. It has been described that exosomes isolated from palmitate-stimulated microglia induced an immature dendritic spine phenotype in hippocampal neurons, probably by enhancing neuroinflammation. Moreover, intranasal administration of neural stem cell-derived exosomes rescued memory in HFD mice [113]. Such results suggest that exosomes may be regulating AMPAR levels, synaptic plasticity, and cognition.

In conclusion, exposure to a saturated HFD causes memory impairment, synaptic dysfunction, and alterations in AMPAR trafficking, most especially in young people and aged. However, unsaturated fatty acids, particularly ω-3, have some benefits at the synaptic level, which could improve cognitive processes in neurodegenerative diseases and counteract the deleterious effects of saturated fatty acids.

## 4. High-Sugar Diet

Findings exclusively focused on HSD are also revealing the specific effects of certain nutrients on cognition. High consumption of (simple) sugars is the other major factor that not only contributes to obesity worldwide, but is also associated with other metabolic diseases like T2D. Yet there are few human studies on HSD effects at the cognitive level [40]. In adolescents, no significant correlation was found between additional sugar intake and memory performance, while negative and positive associations were detected for saturated fatty acids and for ω-3 consumption [86]. Likewise, no association was observed in adults [87], while in middle-aged and older people, a greater intake of total sugars, added sugars, and sugar-sweetened beverages correlated with poorer cognitive function [157]. In all, strong evidence obtained from experiments in animal models enables the conclusion that an HSD, irrespective or not of weight gain, leads to cognitive impairment. Table 3 summarizes animal studies of the effects of an HSD.

### 4.1. Cognition Studies in Animal Models

The form in which added sugar is consumed, as a beverage or as solid food, seems to be independent of its potential to impair cognition. Both sucrose- and fructose-sweetened beverages induced memory deficits in animal models [162,163,167,171], especially in males [160], even when exposure was restricted to 2 h per day [159]. These deficits may be quite permanent, since impairment was reported even one month after sweet solution withdrawal [166]. Similarly, short-term exposure (8 days) to a simple carbohydrate-based diet (mainly sucrose) was enough to damage HPC-dependent memory in a way comparable to a saturated HFD [100]. Especially interesting is the fact that males seem to be more at risk of HSD-induced cognitive deficits, and it is likely that endogenous estrogens may protect memory in females [170]. No significant alterations in working and spatial memory were observed in middle-aged female mice exposed to an HFru solution for 12 weeks [170]. Overall, however, the sex-specific mechanisms of diet-based impairments in cognition are still poorly understood.

### 4.2. Synaptic Function and Neuroplasticity

As far as we are aware, only one study has explored the direct consequences of an HSD on synaptic function using LTP/LTD paradigms. Unexpectedly, no changes in synaptic plasticity measured by the LTP paradigm were detected in CA1-HPC pyramid synapses after 9 weeks of HSu intake; however, LTD was decreased in the temporoammonic pathway, a projection that connects the entorhinal cortex to CA1-HPC [162]. Further research is needed to explore this issue, since learning and memory deficits suggest synaptic dysfunction.

Several studies, nonetheless, have highlighted changes in neuroplasticity-related proteins. Acute exposure to HSu increased HPC GluA1 phosphorylation after 3 days and reduced it after 5–10 days [161], indicating that HSu modifies synaptic levels, while longer-term exposure to HSu increased GluA1 total levels [163]. This effect may be explained by HSu-mediated hyperactivation of mTOR [173] and by aberrant GluA1 synthesis. Regarding BDNF, longer-term exposure to HSu or HFru decreased BDNF levels [158,165] and phosphorylation of its receptors [167], while no changes were detected over shorter periods [100], as happens with a WD.

### 4.3. Insulin Resistance

Daily fructose consumption induced INS resistance in the HPC [167,168,169], while anti-diabetic treatment reversed this effect and enhanced HFru-induced downregulation of BDNF and other synaptic markers in the HPC [165]. Furthermore, added sucrose in a regular diet resulted in a pre-diabetic state, characterized by hyperinsulinemia and peripheral INS resistance [163]; however, this effect has not been studied at brain level.

### 4.4. Other Mechanisms

Mitochondrial dysfunction, reduced neurogenesis, microglia activation, and gut microbiota alterations have been associated with an HSD [100,165,166,167]. Moreover, while HSu exposure produced no effect on inflammation and metabolism [100,163,171], HFru was clearly deleterious [166,168,171]. Neuroinflammation and neuron loss induced by HFru were prevented by short-chain fatty acids, which may exert their protective effects by regulating intestinal homeostasis [164].

Summarizing, with HSD-induced synaptic dysfunction and memory impairment, especially in males and in aging individuals, it would seem that fructose causes more HPC damage than sucrose. However, no study has analyzed HSD outcomes beyond 12 weeks. Since sugar and sweetened beverage intake is higher in adolescents and young adults than in any other age group, it would be interesting to analyze whether an early-life HSD exacerbates age-related impairments in synaptic function and causes late-life cognitive decline.

## 5. Ketogenic Diet

The ketogenic, or keto, diet (KD), high in fats and low in carbohydrates, is based on limiting glucose intake and inducing liver production of ketone bodies (KBs)—acetoacetate, acetone, and β-hydroxybutyrate—and their uptake as an alternative energy source by the brain. The evidence is that KD is effective in reducing or delaying cognitive impairment, in both murine models and human studies. Table 4 summarizes details of human and animal studies of KD and KB supplementation.

### 5.1. Cognition Studies in Humans

The neuroprotective role of ketosis, which also holds for prolonged fasting, has been known for a century, since it reduces the frequency of epileptic seizures. The KD has more recently been tested in humans with AD and diabetes, with limited benefits reported in terms of cognitive and memory enhancement. Regarding AD, ketogenic formulas based on medium-chain triglycerides (MCTs) were associated with higher bloodstream ketone levels, improved short-term cognitive performance [194], and improved longer-term logical memory [192,193]. A very low carbohydrate diet that also increased ketone levels enhanced learning in older adults at high risk of AD [175]. In aged healthy subjects, a single intake of an MCT-based formula was associated with improved performance in working memory tasks [191]. Regarding diabetes, an MCT-based solution prevented hypoglycemia-mediated memory deficits in patients with type 1 diabetes (T1D) [189], while β-hydroxybutyrate (BHB) administration improved working memory in patients with T2D [190]. In contrast, in healthy non-overweight adults, 3 weeks of nutritional ketosis had no effect on performance of several cognitive tasks [174]. In recent years, the KD has gained popularity in the general population, whether self-administrated for memory enhancement or for other conditions, including weight loss, cancer, diabetes, and neurological and psychiatric disorders [201]. However, little is known about the transient symptoms reported during keto-adaptation and the possible side effects of long-term adherence to the KD, so more in-depth studies are needed before it can be recommended to healthy individuals.

### 5.2. Cognition Studies in Animal Models

To better understand KD effects on cognitive performance, we reviewed findings obtained with rodents. In young animals, amygdala-dependent memory remained unaffected by the KD [180], with no differences or even impairments observed in HPC-dependent learning and memory tasks [176,177,178,181,182]. No changes in spatial learning were observed for a high-fat/moderate-protein diet [181], suggesting that protein imbalance may contribute to impaired cognition in some KD experimental models (as also described in [163,164,165,168,169]); this fact could also explain the increased mortality demonstrated for one of the KD paradigms with 75% fat and 8.6% protein (reduced by 10% compared to a control diet) [177]. In adult animals, KD exposure enhanced short-term memory, but no changes were detected for long-term memory [182,184]. Interestingly, an MCT-based formula or cyclic KD (alternated weekly with a control diet) diminished or prevented, respectively, memory decline in aged animals [186,196], as did different KB supplementation paradigms for AD models [188,198,199,200].

### 5.3. Synaptic Function and Neuroplasticity

The KD mechanism of action at the synaptic level remains unknown. However, KD anti-convulsant effects in epileptic patients suggest enhanced inhibition and/or limited excitability in the brain. Indeed, in an epilepsy model, KD conferred protection against focal generation of seizures, although not against seizure spread [202]. In healthy young rodents, the KD did not modify excitability or short-term plasticity, but it did reduce LTP magnitude in the dentate gyrus [179]. However, other studies failed to detect any change in short- or long-term HPC plasticity for the KD [180,183] or KB administration [198,203], supporting the absence of major alterations in cognition. In contrast, in an AD mouse model, the neuroprotective role of KBs improved synaptic plasticity [198], and, according to data from HPC slices, KBs improved recovery after an NMDA-mediated neuronal depression protocol that simulated neuronal injury [204].

One possible explanation for KB improvement of synaptic function may be the ability to induce neuroplasticity-related factors. BHB administration promoted BDNF expression in the HPC [195], while KD-fed mice showed higher HPC levels of GluA1 that correlated with enhanced memory [182]. However, other studies have reported reduced post-KD expression of the *gria1, gria2*, and *gria4* genes (which encode GluA1, 2, and 4, respectively) in the dentate gyrus, as well as of other post-synaptic proteins, while no changes were observed in CA3 [185]; this would suggest subregional-specific modulation of the synaptic function by KD. The fact that reduced AMPAR expression may be explained by KD-mediated inhibition of the mTOR pathway [205] again highlights the importance of distinguishing between the effects of a specific KD (very heterogenous in composition) and KB supplementation.

Since decanoic acid (a saturated fatty acid), but not KBs, has been shown to directly inhibit AMPARs (the mechanism by which seizures are controlled [206]), for AD it has been suggested that MCT supplementation might improve neuronal survival by blocking CP-AMPAR GluA1 homomers [207]. However, further studies are needed to confirm whether AMPAR synaptic level regulation underlies the beneficial effects of MCTs and/or KBs in neurodegeneration by contributing to homeostatic maintenance and delaying memory deficits.

### 5.4. Insulin Resistance

It is well known that KD feeding, low in carbohydrate, reduces blood glucose levels, which is accompanied by a decrease in circulating INS, finally reducing INS resistance [208]. Interestingly, in AD patients, INS infusion, without increasing glucose, improved memory [209]. Moreover, an INS-degrading enzyme was identified as a new player in the removal of INS and Aβ from the brain, evidencing that both peptides were competing for the same enzyme [210]. Consequently, sustained INS levels, associated to INS resistance, enhanced Aβ deposition. In fact, experimentally induced INS resistance boosted β-amyloidosis in AD mouse models (as demonstrated, for example, by [211]). By contrast, BHB treatment of hippocampal neurons reduced Aβ-mediated toxicity in vitro [212], as KD or KBs supplementation did in vivo in mouse models of AD [188,197,198,199,200]. Additionally, KD or MCT supplementation increased INS signaling in the HPC of aged animals [196], specifically in neurons [213], and KD-fed adults showed lower levels of GLUT1, which transports glucose across the BBB, indicating that KBs decreased the dependence for glucose energetic metabolism [184]. It has been suggested that deleterious effects of INS resistance may result from metabolic stress, as neurons gradually lose access to glucose [214]. Therefore, KBs could provide neurons with an alternative fuel, reducing INS resistance-mediated brain damage and improving cognition in neurodegenerative diseases. In summary, KB and MCT diet supplementation shows better results than the KD in protecting synaptic plasticity and delaying cognitive impairment in aging individuals and in neurodegenerative and metabolic diseases. Since there is a lack of evidence of significant cognitive benefits after long-term KD intake in healthy populations, more studies are required to understand whether KBs or MCTs could act as adjuvant or prophylactic therapy to preserve memory in adolescents and adults.

## 6. Paleo(lithic) Diet

The paleo(lithic) diet is the modern interpretation of a dietary pattern based on wild foods and animals, as assumed to be consumed by humans in the Paleolithic era. Based on macronutrient distribution, it is broadly composed of 20–30% protein, 30–45% carbohydrates and 30–40% fat. A more recent definition proposes a new 11-item paleo diet based on no consumption of processed foods; high consumption of fruits, nuts, vegetables, fish, eggs, and lean and unprocessed meats; and minimum content in grains, cereals, and legumes [215]. The absence of grains, cereals, legumes, and other cultivated products, such as olive oil and wine, is the main difference between the paleo diet and the MedDiet [141]. Despite anthropologists holding that our ancestors’ diet heavily influenced their neural complexity and brain size, the benefits of the paleo diet for cognition are poorly studied.

High-protein diets (as an approximation to paleo diet) fed to animal models resulted in no performance differences in learning and memory tasks [216,217,218]. In overweight and T2D human patients, the paleo diet was associated with higher functional brain responses within the HPC [219,220], while in subjects with MetS, the paleo diet improved some cognitive functions and increased serum BDNF levels [221]. While episodic memory performance improved significantly after diet-induced weight loss, no differences were observed between paleo and control diets [219], suggesting that the memory and neuroplasticity effects of dietary interventions may be linked to calorie restriction or INS sensitivity.

In short, the lack of in-depth studies means that no clear conclusions can be drawn as to the effects of the paleo diet at the synaptic and cognitive levels. So far, reported benefits may be improved metabolic health, as with other nutritional interventions. The growing popularity of the paleo diet, especially among young adults and athletes [215], would suggest a need to further study its long-term effects.

## 7. Conclusions

As summarized in this review, the consumption of certain obesogenic food components, such as saturated fatty acids and simple sugars, more so than excessive calorie ingestion, contributes to the decline of both metabolic and cognitive health. Regardless of whether they result in weight gain, their systemic and local effects in the HPC produce neurological damage that leads to progressive learning and memory performance deterioration, enhancing the risk of neurodegenerative diseases, including AD.

Animal models of HFD/HSD intake have provided insights to the impact of fats and sugars on synapsis and their correlation with cognitive decline. However, we found that heterogeneity in experimental approaches (non-specification or variations in the type and proportion of fats and sugars, non-isocaloric control and challenge diets, exposure times, palatability, etc.) prevent the drawing of clear conclusions. That said, we can affirm that an HFD (and especially when composed of saturated and trans fats and cholesterol) and an HSD (sucrose, fructose, and dextrose) produce memory decline at any age. In susceptible life stages—youth and old age—an HFD alone induces other negative effects, on neurodevelopment and maturation, and in exacerbating aging-related decay, respectively; an HSD (fructose more so than sucrose) also has negative effects, even if only consumed over a short period. Although a striking number of studies are carried out exclusively in males, a few studies demonstrate that females are more resistant to diet-mediated memory impairment, due both to a differential choice of calorie sources and the neuroprotective effects of estrogens. More comparative studies of the sexes would allow us to decipher the differential mechanisms at work in men and women, and to identify new cognitive improvement targets.

As far as we are aware, no studies have analyzed whether HFD and HSD effects are additive, but we believe that both impair cognition by simultaneously harming different signaling pathways with the same outcomes. Figure 3 depicts the main hallmarks of HFD/HSD-mediated HPC damage, showing the correlation with memory and learning loss, even after short-term exposure. This is especially noticeable in reduced dendritic arborization, a widened synaptic cleft, narrowed postsynaptic density, and decreased PSD95 levels, while changes in other classically analyzed pre/post-synaptic markers are less consistent. Regarding the neuroplasticity marker BDNF, we suggest that reduced levels are a medium-term consequence of synaptic dysfunction, rather than a direct cause. The decrease in BDNF may also reduce GluA1 synthesis through the mTOR pathway. Furthermore, correlating perfectly with cognitive impairment is lower activity-dependent synaptic plasticity, exemplified by a decrease in LTP, whereas LTD is not always affected.

We believe that all those alterations converge in modifying synaptic AMPAR levels. Over the short/medium-term, AMPAR total and synaptic levels are reduced, and palmitoylation and phosphorylation (both regulatory processes of GluA1 trafficking to the PM) are increased and decreased, respectively. However, long-term nutrition with HFD may enhance AMPAR synaptic levels, contributing to excessive Ca^2+^ influx and, ultimately, producing synaptic dysfunction as described for brain diseases.

Therapeutic approaches and nutritional interventions are available that slow cognitive decline associated with aging and with neurodegenerative and metabolic diseases. Metformin can partially reverse HFD/HSD-mediated memory decline, which is especially interesting for patients with MetS, while ω-3, MCTs, or KBs (preferable to the KD) diet supplementation seems to improve cognitive performance in patients with AD and diabetes. At the molecular level, the MedDiet, with high ω-3 content (present in fish, seafood, soybean oil, nuts, and seeds), or diet supplementation with ω-3 blocks synaptic damage linked to aging and neurodegeneration and slows cognitive decline. Nutrients seem to mainly affect younger and aged people, so long-term nutritional health programs addressed to these age groups may produce the greatest benefits in terms of synaptic plasticity, learning, and memory. A final interesting note is that ω-3 fatty acids can reverse most of the molecular deleterious effects of saturated fatty acids in HPC synapsis; this is good news for the nutrition field, since well-balanced consumption of foods rich in saturated fats (cheese, butter, meat, etc.) combined with foods rich in unsaturated fats (vegetables oils, fish, seeds, etc.) does not appear to harm cognitive health.

## Figures and Tables

**Figure 1 nutrients-14-04137-f001:**
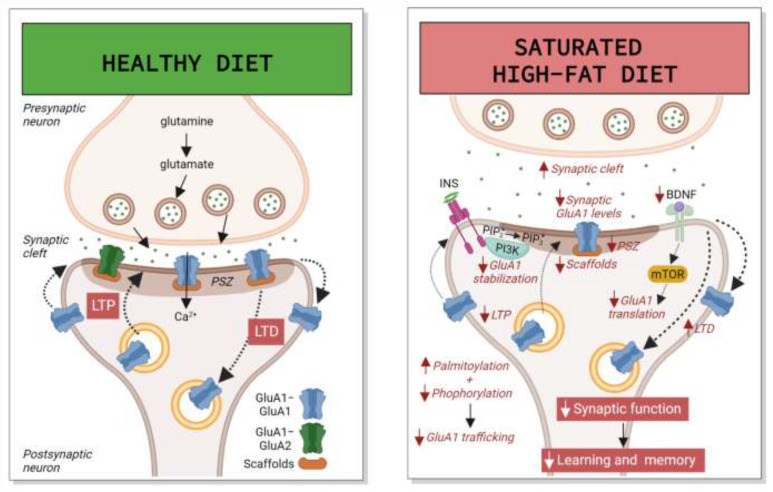
Mixed and saturated high-fat diet (HFD) deregulation of AMPARs in the hippocampus (HPC). In normal conditions, glutamate release induces activation of ionotropic α-amino-3-hydroxy-5-methyl-4-isoxazolepropionic acid receptors (AMPARs), both GluA1-GluA1 homomers and GluA1-GluA2 heteromers, which mediate fast excitatory synaptic transmission. Under neuroplasticity paradigms like long-term potentiation (LTP) and long-term depression (LTD), AMPAR levels in the post-synaptic zone (PSZ) increase or decrease by exocytosis or endocytosis, respectively, or through changes in their lateral diffusion, mainly controlled by phosphorylation and dephosphorylation. An intake of saturated HFD decreases LTP, enhances LTD, and reduces total GluA2 and synaptic GluA1 levels. This decrease in GluA1 may be a consequence of increased palmytoilation mediated by palmitic acid (PA) and insulin (INS) and decreased phosphorylation, and by a decline in the levels of auxiliary and scaffold proteins. All these processes may impair trafficking of GluA1 towards the plasma membrane (PM). Moreover, PA-induced phosphorylation of PI(4,5)P_2_ (also called PIP_2_) by phosphoinositide 3-kinase (PI3K), a protein kinase activated by INS, may compromise surface GluA1 stabilization, and decayed GluA1 synthesis, mediated by brain-derived neurotrophic factor (BDNF) through the mammalian target of rapamycin (mTOR), may downregulate both total and synaptic levels. Altogether, the outcome is synaptic dysfunction and learning and memory impairment. Other figure abbreviations are as follows: Ca^2+^: calcium ion; PIP_2_: phosphatidylinositol (4,5)-bisphosphate; PIP_3_: phosphatidylinositol (3,4,5)-trisphosphate. Image created with Biorender.com.

**Figure 2 nutrients-14-04137-f002:**
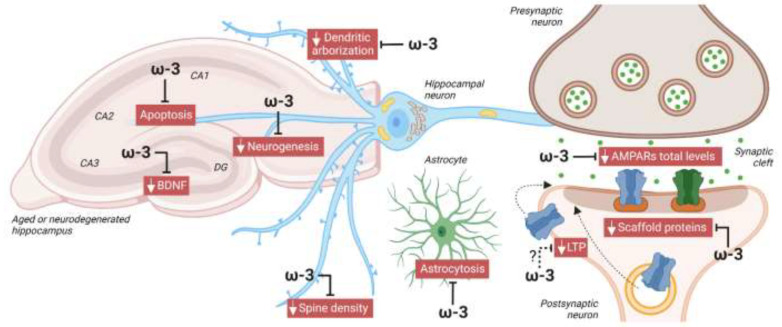
Protective role of omega 3 (ω-3) against neurodegeneration. In animal models of aging or neurodegeneration, diet supplementation with ω-3 reduced apoptosis and astrocytosis, and enhanced neurogenesis, dendritic arborization, spine density, and levels of BDNF, AMPARs, and scaffold proteins, like postsynaptic density protein 95 (PSD95), in the HPC. The fact that ω-3 deficiency reduced LTP in aged mouse CA1-HPC suggests that ω-3 could also improve activity-dependent synaptic plasticity, contributing to its protective role in cognition. Image created with BioRender.com.

**Figure 3 nutrients-14-04137-f003:**
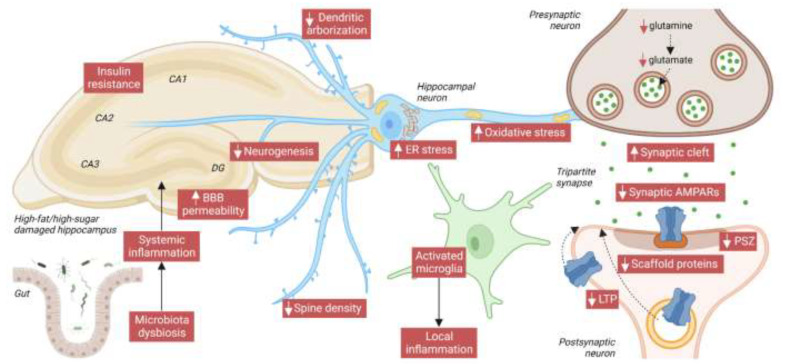
Main hallmarks of HFD/high-sugar-(HSD)-mediated HPC damage. At the systemic level, both fat and sugar produce gut microbiota dysbiosis, which can exacerbate inflammation in the brain by enhancing blood-brain barrier (BBB) permeability. BBB damage, in turn, contributes to deregulating the transport of circulating orexigenic and anorexigenic hormones. In the HPC, local release of neuroinflammatory factors by activated microglia exacerbates neuronal damage. INS resistance, increased endoplasmic reticulum (ER), and oxidative stress also compromise neuronal function. In HPC neurons, HFD/HSD intake reduces dendritic arborization, spine density, and the PSZ, and widens the synaptic cleft. The glutamate neurotransmitter and its precursor, glutamine, are reduced, and also reduced over the short/medium-term are levels of AMPARs and of its scaffold protein PSD95. Finally, impaired activity-dependent synaptic plasticity (especially LTP) proves that AMPAR trafficking to the PM is also compromised by an HFD/HSD. Image created with Biorender.com.

**Table 1 nutrients-14-04137-t001:** WD effects on synaptic function and cognition in animals.

Exp. Approach	Species, Sex, Age	Model	Learning/Memory	Synaptic Function, Neuroplasticity	Other Pathways	Refs.
Short periods (≤2 months)						
Young animals	Rat (♀; 3 w)	CAFD (F/sucrose: 45%) & HFru solution (11%) or HFru solution alone vs. CD (F: 13.4% & C: 56.7%) for 5 w. Reversion: 5 w with CD	Impaired memory (novel object in context) at 5 w for CAFD and HFru solution. No reversion in CAFD group	-	Gut dysbiosis (before and after reversion)	[44]
	Rat (♂; 6 w)	HFD-HDextrose (F: 41.7% & C: 36.7%) vs. CD (F: 13.5% & C: 58%) for 11 w	Impaired memory (NOR; no changes with MWM)	↓ dendritic arborization in HPC neurons and ↑ in entorhinal cortex neurons	↑ TNFα levels in blood	[45]
	Rat (♂; 6 w)	CAFD (cakes, biscuits & a protein source) & HSu solution (10%) vs. CD for 6 w	Impaired memory (NOR and NLR)	No changes in BDNF, TrkB and synapsin in HPC	↑ inflammation and gut dysbiosis	[46]
	Rat (♀; 2 m)	HFD (F:39%) & refined sugar (40%) vs. CD (F: 13% & complex C: 59%) for 2 m	Impaired memory (MWM)	↓ BDNF, phosphosynapsin I and phosphoCREB	-	[47]
Adult animals	Rat (♂ and ♀; 9–10 w)	HFD (F: 60% & C: 20%) HFru solution (11%) vs. CD (F: 13% & C: 62%) for 6 w	Impaired hippocampal-dependent memory (NLR) in ♂ (no changes in ♀)	-	-	[48]
	Rat (♂; adult)	CAFD (F: 45% & C: 50%) & HSu solution (10%) vs. CD (F: 15%, & C: 59%) for 5, 11 & 20 days	Impaired hippocampal-dependent memory (NLR; no changes in NOR)	No changes in BDNF	↑ inflammation in HPC at 20 d	[49]
	Rat (♂)	CAFD (CD supplemented with cakes, biscuits & protein source) & HSu solution (10%) vs. CD for 5 w	Impaired memory (NLR)	-	Gut dysbiosis	[50]
	Rat (♂; adult)	HFD-HC (F: 25%, C: 44% & P: 18%) vs. CD (F: 5%, C: 62% & P: 18) for 6 w	Impaired short-term and long-term memory (RAWM)	No changes in BDNF	↑ oxidative stress in HPC	[51]
	Rat (♂)	HFD (F: 40%, P: 5% & C: 15%) & HSu solution (40%) vs. CD (F: 15%, P: 25% & C: 55%) for 6 w. Reversion: 3 w with HFD-HSu and 3 w with CD and training	Impaired memory (NLR). Reversed by CD and training	-	-	[52]
	Rat (♂)	HFD (SFAs & MUFAs; 38%) & refined sugar (38%) vs. CD (F: 6% & sugar: 4.1%) for 8 w	Impaired memory (NOR) and learning (MWM)	↓ GluA3 levels in dorsal HPC and altered levels in synaptic plasticity markers	Altered levels in energy metabolism markers (by proteomic analysis)	[53]
Long periods (>2 months)						
Young animals	Rat (♂; 21 days)	HFD (F: 29%, sucrose: 34% & cholesterol 1.25%) & HGlucose-HFru solution (55%/45%) vs. CD (F: 6% & C: 44%) for 8 m	-	↓ PSD95 and BDNF, ↓ LTP in HPC	Gut dysbiosis, ↑ inflammation and microglia activation in HPC	[54]
	Rat (♀; 1 m)	HFD (30% lard & 66% sucrose) & HSu solution (30%) vs. CD (F: 3%, C: 61% & P: 19%) for 24 m	Impaired memory (NOR)	↓ TrKB, ↓ LTP in HPC	No changes in neurogenesis	[55]
	Rat (♂; 5–7 w)	HFD & HSu solution (5%) vs. CD for 4 m. Reversion: 3 m with bioactive food	Impaired spatial and working memory (T-maze and NOR). Reversed by bioactive foods	-	Gut dysbiosis. Reversed by bioactive foods	[56]
	Rat (♂; 6 w)	HFD-HFru (F: 30% & fructose: 15%) vs. CD for 6 m	Impaired learning (MWM)	-	↑ BBB permeability, neurodegeneration and microglia activation	[57]
	Rat (♂; 6 w)	HFD-HFru (saturated F: 45% & fructose: 20%) vs. CD for 11 w	Impaired memory (NOR)	-	↓ IGF1 and ↑ oxidative stress	[58]
	Mouse (♂; 6 w)	HFD-HSu (F: 60% & sucrose: 7%) vs. CD (F: 17%) for 13 w	Impaired memory (NOR)	↓ GluA1, BDNF, phosphoCREB, TrkB in HPC and PFC	↓ neurogenesis	[59]
	Mouse (♂; 7 w)	HFD (F: 40% & C: 20%) & HSu solution vs. CD (F: 12% & C: 67%) for 14 w	-	↓ PSD95 and ↑ phosphoTau in brain (no changes in synaptophysin)	↓ GLUT1/3, ↑ ER stress and inflammation responses and INS resistance in brain	[60]
	Rat (♂; 7 w)	CAFD (CD with cookies, cakes & biscuits) vs. CD once per day, 5 days per week for 5 m	-	↑ BDNF and TrkB and ↓ phosphoTrkB in PFC. No changes in HPC	Redox imbalance	[61]
	Rat (♀; 8–10 w)	HFD (F: 40%, C: 45% & P 15%) & HFru solution (15%) vs. CD (F: 6%, C: 64% & P: 25%) for 12, 16 & 24 w	Impaired memory (MWM) at 16 and 24 w (no changes at 12 w)	-	↑ oxidative stress and reduced antioxidant levels in HPC and CTX	[43]
	Rat (♂; 2 m)	HFD-HGlucose (F: 40%) vs. CD (F: 13%) for 3 m	Impaired learning (nonspatial discrimination learning problem)	-	↑ BBB permeability in HPC	[62]
	Rat (♀; 2 m)	HFD (SFAs & MUFAs: 39%) & refined sugar (40%) vs. CD (F: 13% & C: 59%) for 1, 2 & 6 m or 2 y	Impaired learning and memory (MWM) at 1 and 2 m	↓ BDNF and synapsin I in HPC	-	[63]
	Rat (♂; 2 m)	HFD (high-lard or high-olive oil) & HSu solution vs. CD for 10 w	No changes in spatial memory (Y-maze)	↓ GluN2A in high-lard-HSu (no changes in high-olive oil-HSu) in CTX	-	[42]
	Rat (♂; 2 m)	HFD-HSu & HFru corn syrup solution (20%) vs. CD for 8 m	Impaired learning (MWM)	↓ dendritic spine density, ↓ LTP and ↓ BDNF levels in CA1-HPC	-	[64]
	Mouse (♀ and ♂; 10 w)	HFD (F: 60%) & HSu solution (20%) vs. CD (F: 10%) for 4 or 6 m. Reversion: 8 w with CD	Impaired memory (NOR and NLR. No change in working memory (Y-maze). Recovered after 8 w of CD		↑ microglia activation (no inflammation and neuronal loss). Recovered after 8 w of CD	[65]
	Guinea pigs (♀; 10 w)	HFD-HSu (F: 20% & sucrose: 15%) vs. CD (F: 4% & sucrose: 0%) for 7 m	-	↓ BDNF levels in HPC	-	[66]
	Rat (♂; adolescent)	HFD-HDextrose (SFAs: 41.7%) vs. CD (F: 13.4%) for 10 w	Impaired memory (NOR)	-	-	[67]
Adult animals	Mouse (♂; 3 m)	HFD (F: 45%) & HFru solution (10%) vs. CD for 10 w	Impaired memory (MWM)	↓ PSD95 and SNAP25 in HPC	INS resistance, ↑ microglia activation and inflammation in brain	[68]
	Rat (♂)	HFD-HDextrose (F: 38%, P: 24%, C: 18% & dextrose: 20%) vs. CD (F: 18%, P: 24%, C: 58%) for 10, 40 & 90 days	Impaired learning at 10 and 90 d, no changes at 40 d (Y-shaped maze)	-	↑ BBB permeability in HPC (only in obese rats)	[41]
	Mouse (♂; 3 m)	HFD-HFru (F: 48%, fructose: 33% & P: 19%) or HFD (F: 48%, C: 33% & P: 19%) vs. CD for 14 w	Impaired memory (NOR and NLR). HF-HFru more affected than HFD	↓ glutamate and glutamine in HPC (no changes in GABA)	-	[69]
	Rat (♂)	HFD-HDextrose (F: 40%, P: 21% & C: 38%) or HFD-HSu (F: 40%, P: 21% & C: 38%) vs. CD (F: 12%, P: 28% & C: 59%) for 3 m	Impaired learning (nonspatial Pavlovian discrimination and reversal learning problem)	↓ BDNF in prefrontal CTX and HPC with HF-HDextrose (no changes with HF-HSu)	-	[70]
	Rat (♂ and ♀)	HFD-HFru (CD: 60%, fructose: 30% & pork fat: 10%) vs. CD for 12 w	Impaired learning (MWM and passive avoidance test). ♂ more affected than ♀	-	↑ oxidative stress	[71]
Short vs. long periods						
Young animals	Mouse (♂; 6 w)	HFD-HFru (30% lard, 0.5% cholesterol and 15% fructose, all in weight/weight) vs. CD for 4 or 24 w	Impaired learning (MWM) at 14 w (no changes at 4 w)	-	↑ BBB permeability and astrocytocis	[72]

WD effects in animals were classified according to experimental approach, taking into account exposure time, short (≤2 months) or long (>2 months) periods, sex, and age classified as young and adult (≤2 months, >2 months). Search terms used were as follows: “western diet” OR “high fat” AND “sugar”, “sucrose”, “fructose” OR “dextrose” AND “memory”, “learning”, “synaptic plasticity” OR “AMPAR”. Only studies with rodents were included. Excluded were experimental models of maternal exposure, streptozotocin-treated animals, and pathological conditions. Percentage fats, carbohydrates, and proteins are relative to total energy unless otherwise indicated. Solution percentages are expressed as weight/volume unless otherwise indicated. Table symbols and abbreviations are as follows: ♀: female; ♂: male; ↑: increased; ↓: decreased; BBB: blood-brain barrier; BDNF: brain-derived neurotrophic factor; C: carbohydrates; CAFD: cafeteria diet; CD: control diet; CREB: cAMP response element-binding protein; CTX: cortex; ER: endoplasmic reticulum; F: fats; FC: fear conditioning; GABA: gamma-aminobutyric acid; GluA: AMPA receptor; GluN: NMDA receptor; GLUT: glucose transporter; h: hours; HC: high carbohydrate; HDextrose: high dextrose; HFD: high-fat diet; HFru: high fructose; HGlucose: high glucose; HPC: hippocampus; HSD: high-sugar diet; HSu: high sucrose; INS: insulin; IGF1: insulin-like growth factor 1; LTD: long-term depression; LTP: long-term potentiation; m: months; MUFAs: monounsaturated fatty acids; MWM: Morris water maze; NLR: novel location recognition; NOR: novel object recognition; P: protein; PFC: prefrontal cortex; phospho: phosphorylated; PSD95: postsynaptic density protein 95; PUFAs, polyunsaturated fatty acids; RAWM: radial arm water maze; SFAs: saturated fatty acids; SNAP25: synaptosomal-associated protein 25; SOLF: saturated oil-enriched food; TNFα: tumor necrosis factor; TrkB: tropomyosin receptor kinase B; UOLF: unsaturated oil-enriched food; vs.: versus; and w: weeks.

**Table 2 nutrients-14-04137-t002:** Mixed or saturated HFD effects on synaptic function and cognition in animals.

Exp. Approach	Species, Sex, Age	Model	Learning/Memory	Synaptic Function, Neuroplasticity	Other Pathways	Refs.
Days (≤2 weeks)						
Young animals	Rat (♂; 3 w & 2 m)	Saturated HFD (F: 60% & C: 20%) vs. CD (C: 35%) for 7 days	Impaired long-term memory (NLR) in young animals but improved in old ones. No changes in short-term memory	↓ LTP in CA1-HPC in young animals, but ↑ in old ones	↑ glucocorticoids release in young animals	[103]
	Mouse (♂; 5 w)	HFD (F: 45% & C: 35%) vs. CD (F: 18% & C: 58%) for 2 days	-	↓ LTP in HPC. No changes in LTD	-	[104]
	Mouse (♂; 6 w)	HFD (F: 60% & C: 27%) vs. CD (F: 10% & C: 70%) for 1 to 7 days	Impaired memory (NOR) at 3–7 days	↓ synaptophysin at 7 days in HPC. No changes at 3 days	↑ BBB permeability at 1–3 days (no changes at 4–6 days) and inflammation at 2 days (no changes at 1 or 3–7 days)	[105]
	Mouse (♂; 6 w)	HFD (F: 60% & C: 21%) vs. CD (F: 13% & C: 67%) for 7 days	-	↓ BDNF and dendritic tree in HPC	-	[106]
Adult animals	Rat (♂)	Saturated HFD (F: 48% & C: 37%) or PUFA HFD (F: 46% & C: 37%) vs. CD (F: 21% & C: 56%) for 2 w	Impaired memory (NLR) with saturated. No changes with PUFA	No changes in BDNF in HPC	Gut dysbiosis. No changes in inflammation in HPC	[100]
	Mouse (♂; 10 w)	HFD (F: 30% by weight) vs. CD (F: 5% by weight) for 7 days	Impaired memory (NOR, Y-maze & temporal order memory test)	↓ PSD95, BDNF, thickened of post synaptic density, & ↑ width of synaptic cleft in HPC & PFC	Gut dysbiosis, ↑ microglia activation, & inflammation in HPC & PFC	[93]
	Mouse (♂; 12 w)	Saturated HFD (F: 60%) vs. CD (F: 10%) in animals for 3 days, 1 or 2 w	-	-	Alterations in metabolism, cell stress, inflammation, cell signaling & cytoskeleton	[107]
Young vs. aged animals	Rat (♂; 3 m & 24 m)	Mixed HFD (F: 60.3% & C: 21.3%) vs. CD (F:17% & C: 54%) for 3 days	Impaired long-term memory (FC & MWM) in aged animals. No changes in short-term memory or in young animals	-	↑ microglia activation & inflammation in aged animals in HPC & amygdala	[92]
	Rat (♂; 3 m & 24 m)	Mixed HFD (F: 60.3% & C: 21.3%) vs. CD (F:17% & C: 54%) for 3 days	-	-	↑ inflammation in microglia of young & old animals in HPC & amygdala	[108]
Short periods (>2 weeks and ≤2 months)						
Young animals	Rat (♂; 3 w)	Mixed HFD (F: 42% & C: 25%) or CD (F: 4% & C: 50%) for 7 w	-	↓ GluA2, PSD95, synaptophysin, & TrKB receptor in HPC. No changes in BDNF	↓ glucocorticoid receptor in HPC	[109]
	Mouse (♂; 3 w)	HFD (F: 60%) vs. CD (F: 6.55%) for 8 w	-	↓ response to leptin induction of AMPAR-mediated synaptic transmission in HPC neurons	-	[110]
	Mouse (♂; 3 w)	HFD (F: 21.2% & C: 22.5% by weight) & CD (F:3.6% & C: 28.8% by weight) for 6 w	Impaired memory (NOR)	Synaptic loss in CA1-HPC neurons	↓ neurogenesis and ↑ inflammation in HPC	[111]
	Mouse (♂; 8 w)	HFD (F: 60% & C: 20%) calorically matched or ad libitum vs. CD (F: 13% & C: 58%) for 3 w.	-	↓ spine density in PFC in both HFD protocols	-	[112]
	Mouse (♂; 4 w)	Saturated HFD (F: 60%) vs. CD (F: 6.55%) for 6–7 w	Impaired memory (MWM)	↓ LTP in HPC. ↓ GluA1 phosphorylation & ↑ palmitoylation in HPC	INS resistance in HPC	[32]
	Mouse (♂; 4 w)	Saturated HFD (F: 60%) vs. CD (F: 6.55%) for 6–7 w	Impaired memory (NOR & NLR)	↓ BDNF, phosphoTRKB, phosphoCREB & glutamate metabotropic receptors in HPC	-	[113]
	Mouse (♂; 5 & 8 w)	SOLF (60% CD + 40% saturated oil-enriched food) or UOLF (60% CD + 40% unsaturated oil-enriched food) for 8 w	Impaired memory (Y-maze) by SOLF	↓ GluN2A & 2B by SOLF/UOLF in young animals (no changes in old). ↓ LTP & ↑ LTD by SOLF in young animals	-	[102]
	Rat (♂; 6 w)	HFD (F: 40%) vs. CD for 6 w	Impaired learning & memory (MWM)	-	↑ ER stress & INS resistance in HPC	[114]
	Rat (♂; 7 w)	HFD (F: 40% & C: 40%) vs. CD (F: 12.5% & C: 62.9%) for 8 w	-	↓ BDNF in PFC	↑ ER stress in PFC	[115]
	Rat (♂; 2 m)	HFD (5000 kcal/kg) vs. CD (3600 kcal/kg) for 8 w	Impaired learning & memory (MWM)	↓ synaptotagmin1 & synapsin 1	-	[116]
	Mouse (♂; 6–8 w)	HFD (F: 21.2% & C: 61.3% by weight) vs. CD (F: 7.5% & C: 75.1% by weight) for 4 or 7 w	Impaired long-term memory (FC). No changes in short-term memory (NLR)	↓ LTP in CA1-HPC. ↓ cfos, synaptophysin, CaMKII & IV, calcineurin A in HPC. No changes in BDNF	↑ oxidative stress	[117]
	Mouse (♂; 8 w)	HFD (F: 45%) vs. CD (F: 10%) for 2 m	Impaired short-term memory (T-maze)	-	INS resistance in brain	[118]
Adult animals	Rat (♂; adult)	HFD (F: 58% & C: 17%) vs. CD for 5 w	Impaired learning & memory (MWM)	-	-	[119]
	Rat (♂)	HFD (F: 20% & C: 48% by weight) vs. CD (F: 5% & C: 47% by weight) for 8 w	-	No changes in LTP in DG-HPC	-	[120]
	Rat	Mixed HFD (F: 39%) vs. CD (F: 13%) for 2 m	-	-	↑ oxidative stress	[121]
	Rat (♂; 16 m)	Saturated HFD (2 % cholesterol + 10 % trans coconut oil) vs. soybean oil (12%) for 8 w	Impaired memory (water radial arm maze)	Dendritic loss in HPC	↑ microglia activation & inflammation in HPC	[101]
	Mouse (♂)	Mixed HFD (F: 45% & C: 45%) vs. CD (F: 10% & C: 70%) for 8 w	No changes in memory (Y-maze & NOR)	-	-	[122]
Long periods (>2 months)						
Young animals	Mouse (♂; 4 w)	Mixed HFD (F: 60% & C: 20%) vs. SD (F: 10% & C: 70%) for 14 w	Impaired memory (Y-maze & MWM)	↓ BDNF in HPC & CTX	Altered antioxidant defense, ↑ oxidative stress, inflammation & INS resistance in HPC & CTX	[123]
	Rat (♂; 5–6 w)	HFD (F: 59.28%) vs. CD (F: 19.77%) for 16 w	Impaired memory (MWM)	↓ spine density & LTP in HPC	INS resistance & ↓ mitochondrial function in brain	[124]
	Rat (♂; 6 w)	Mixed HFD (F: 60% & C: 20%) vs. CD (F: 13% & C: 58%) for 6 m	Impaired learning (FC)	↑ surface GluA1 in HPC. No changes in total levels or GluN2B	-	[125]
	Mouse (♂; 6 w)	Mixed HFD (F: 60%) vs. CD (F: 12.6%) for 16 w	Impaired memory (Y-maze)	↓ GluN1/2A, GluA1, PSD95 & synaptophysin. No changes in GluN2B or GluA2	-	[126]
	Mouse (♂; 6 w)	HFD (F: 22% by weight) vs. CD (F: 6% by weight) for 16 w	-	-	↑ inflammation in HPC & CTX & gliosis in CTX	[127]
	Rat (♂; 7 w)	Mixed HFD (F: 45%) vs. CD (F: 6%) for 17 w	-	↓ PSD95. No changes in synaptophysin	-	[128]
	Mouse (♂; 8 w)	Saturated HFD (F: 59% & C: 26%) vs. CD (F: 11% & C: 59%) for 8, 16, 24 & 28 w	No changes in short-term (Y-maze), long-term & learning (MWM). Impaired cognitive flexibility	-	No changes in microglia activation in HPC & CTX	[129]
	Mouse (♂; 8 w)	Mixed HFD (F: 60% & C: 20%) vs. CD (F: 10% & C 70%) for 46 w	Impaired memory (MWM, NOR & Y-maze)	↓ branching, spine density, PSD95, spinophilin, & synaptophysin in HPC	↑ microglia activation, inflammation & iNOS in HPC	[130]
	Mouse (♂; 8 w)	Mixed HFD (F: 60%) vs. CD for 16 w	Impaired learning & memory (MWM)	-	↑ inflammation & INS resistance in HPC & CTX	[131]
	Mouse (♂; 8–10 w)	Mixed HFD (F: 45%) vs. CD (F: 10% & C: 60%) for 13 w	-	↓ synaptophysin in HPC	↑ oxidative stress & INS resistance, & ↓ neurogenesis in HPC	[132]
	Mouse (♀; 2 m)	Mixed HFD (F: 60%) vs. CD (F: 10%) for 4 m. Intervention: 16 m with CD	Impaired learning & memory (MWM & FC)	↓ BDNF in HPC	No changes in inflammation or mitochondrial function in HPC	[95]
Adult animals	Mouse (♂)	HFD (F: 45%) vs. CD (F: 10%) for 17 w	Impaired memory (T-maze)	↓ LTP in HPC	-	[96]
	Mouse (♂; 9 w)	HFD (F: 60% by weight) vs. CD (F: 5% by weight) for 13–15 w	Impaired memory (NOR & NLR)	↓ BDNF, synaptophysin, & PSD95 in HPC & PFC	Gut dysbiosis, ↑ inflammation, & microglia activation in HPC & PFC	[97]
	Mouse (♂; 9 w)	Mixed HFD (F: 60% & C: 20%) vs. CD (F: 10% & C: 70%) for 24 w	Impaired memory (MWM & Y-maze)	↓ PSD95 & SNAP23 in HPC & CTX	↑ inflammation, microglia activation, oxidative stress, INS resistance & Aβ in HPC & CTX	[98]
	Mouse (♂; 12 w)	HFD (F: 55%) vs. CD (F: 13%) for 15 w	Impaired memory (NOR & NLR)	↓ PSD95, synaptophysin, thickened of post synaptic density, & ↑ width of synaptic cleft in CA1-HPC	Gut dysbiosis, ↑ inflammation, microglia activation, & INS resistance in CA1-HPC	[99]

Mixed or saturated HFD effects were classified according to experimental approach, taking into account exposure time, days (≤2 weeks), short (>2 weeks and ≤2 months) or long (>2 months) periods, sex, and age classified as young and adult (≤2 months, >2 months). Search terms used were as follows: “high fat diet” AND “learning”, “memory”, “synaptic plasticity” OR “AMPAR”. Only studies with rodents were included. Excluded were experimental models of maternal exposure. Percentage fats, carbohydrates, and proteins are relative to total energy unless otherwise indicated. Table abbreviations (used for the first time in this table) are as follows: Aβ: amyloid-β; DG: dentate gyrus; iNOS: inducible nitric oxide synthase; and SNAP23: synaptosome-associated protein 23.

**Table 3 nutrients-14-04137-t003:** HSD effects on synaptic function and cognition in animals.

Exp. Approach	Species, Sex, Age	Diet Model	Learning/Memory	Synaptic Function, Neuroplasticity	Other Pathways	Refs.
High sucrose						
Young animals	Rat (♂; 3 w)	HSu diet vs. CD for 8 w	-	↓ BDNF and synaptophysin in brain	-	[158]
	Rat (♂; 3 and 8 w)	HSu solution (10%) 2 h/day vs. 0.1% sodium saccharin solution for 4 w	Impaired learning and memory (MWM)	-	-	[159]
	Rat (♂ and ♀; 4 w)	HSu solution (10%) 2 h/day vs. water for 2 & 4 w	Impaired memory (NLR; more deficits in ♂)	-	-	[160]
	Rat (♂; 8 w)	HSu solution (32%) vs. water for 3, 5 or 10 days	-	↑ GluA1 phosphorylation in dorsal HPC at 3 days and ↓ at 5 and 10 days	-	[161]
	Rat (♂; 12 w)	HSu solution (35%) vs. water for 9 w	Impaired memory (NLR and NOR)	↓ LTD in CA1-HPC (no changes in LTP)	No changes in metabolic profile in HPC	[162]
Adult animals	Rat (♂; 4 m)	HSu solution (35%) vs. water for 9 w	Impaired memory (MWM and Y-maze)	↑ GluA1 and GluN1 protein levels in HPC	No changes in oxidative stress or inflammation in HPC	[163]
High fructose						
Young animals	Mouse (♂; 5 w)	HFru (35%) vs. CD for 8 w	-	-	Microglia activation, ↑ inflammation, ↓ neurogenesis, and neuronal loss in HPC	[164]
	Rat (♂; 6 w)	HFru (60%) vs. CD for 12 w	-	↓ BDNF and PSD95 in HPC	INS resistance and microglia activation, and ↓ neurogenesis in HPC	[165]
	Rat (♂; 6 w)	HFru solution (10%) vs. water for 12 w	Impaired spatial memory (Barnes maze)	-	Astrocytosis, ↓ neurogenesis, ↑ inflammation, and in HPC and PFC	[166]
	Rat (♂; 2 m)	HFru solution (15%) vs. water for 8 w	Impaired memory (Barnes maze)	↓ phosphoTrKB and synaptophysin. No changes in BDNF in HPC	Alterations in metabolism, mitochondrial function and INS resistance in HPC	[167]
	Rat (♂; 12 w)	HFru solution (10 % or 60%) vs. water for 9 w	-	-	INS resistance and ↓ inflammation in HPC with 10% of HFru. No changes in Ins and ↑ inflammation with 60% of HFru	[168]
Adult animals	Rat (♂)	HFru solution (15%) vs. water for 6 w	Impaired memory (Barnes maze)	-	INS resistance in HPC	[169]
	Mouse (♀; 9 m)	HFru solution (10%) vs. water for 12 w	No changes in spatial memory (Y maze)	-	↓ antioxidant defense in PFC	[170]
High sucrose vs. high fructose						
Young animals	Rat (♂; 4 and 9 w)	HSu or HFru solution (11%) vs. water for 30 days	Impaired memory in young animals with HFru (Barnes maze). No changes in adults or with HSu	-	↑ inflammation in young animals with HFru. No changes in adults or with HSu	[171]
Simple carbohydrates						
Adult animals	Rat (♂)	HC (simple C: 30%) vs. CD (simple C: 16%) for 8 days	Impaired memory (NLR)	No changes in BDNF in HPC	Gut dysbiosis. No changes in inflammation in HPC	[100]
	Mouse (♂; 22 w)	HC (simple C: 36% weight) vs. CD (simple C: 12%) for 10 w	-	-	↑ expression of neurodegeneration genes, inflammation, mitochondrial function, and oxidation in HPC	[172]

HSD effects were classified according to experimental approach, taking into account sugar type (high sucrose, high fructose, high sucrose vs. high fructose, or simple carbohydrates), sex, and age classified as young or adult (≤2 months, >2 months). Search terms used were as follows: “high sucrose” OR “high fructose” AND “learning”, “memory”, “synaptic plasticity” OR “AMPAR”. Only studies with rodents were included. Excluded were experimental models of maternal exposure and pathological conditions. Carbohydrate percentages are relative to total energy unless otherwise indicated.

**Table 4 nutrients-14-04137-t004:** KD and KB supplementation effects on synaptic function and cognition in humans and animals.

Exp. Approach	Species, Sex, Age	Diet Model	Learning/Memory	Synaptic Function, Neuroplasticity	Other Pathways	Refs.
Ketogenic diet						
Humans	Human (♂ and ♀; 18–40 y)	KD (F: 60%, C: 15% & P: 25%) vs. CD (F: 20%, C: 55% & P 25%) for 3 w	No changes in cognitive performance	-	-	[174]
	Human (♂ and ♀; ~70 y)	KD (C: 5–10%) vs. CD (C: 50%) for 6 w	Improved paired associate learning	-	-	[175]
Young animals	Mouse (♂; 3 w)	KD (F: 90.5%, C: 0.3% & P: 9.1%) vs. CD (F: 13%, C: 67% & P: 20%) for 2 w	No differences in learning (Hebb Williams Maze) or memory (passive avoidance test)	-	-	[176]
	Mouse (♂; 3 w)	KD (F: 75.1%, C: 3.2% & P: 8.6%) vs. CD (F: 7.1%, C: 63.2% & P: 18.3%) for 5.5 w	Impaired learning (MWM) but no differences in memory (NOR)	-	-	[177]
	Rat (♂; 3 w)	KD (F: 78.8%, C: 0.8% & P: 9.5%) vs. CD (F: 10%, C: 49% & P: 23.4%) for 1 m	Impaired learning and memory (MWM)	-	-	[178]
	Rat (♂; 3 w)	KD (F: 87% & C + P: 13%) vs. CD for 3 w	-	No changes in short-term plasticity but ↓ LTP magnitude	-	[179]
	Rat (3 w)	KD (F: 92%, C: 3% & P: 5%) vs. CD (F: 12%, C: 65% & P: 24%) for 2–3 w	No changes in memory (FC)	No changes in short-term or long-term plasticity	-	[180]
	Rat (♂; 4 w)	KD (F: 69%, P: 24% & C: 0%) vs. CD (F: 12%, P: 23% & C: 54%) for 6 w	No changes in learning (MWM)	-	↑ transitory glia activation in CA3-HPC at 1 w (no changes at 6 w)	[181]
	Rat (♂; 7–8 w)	KD (F: 90%, P: 10% & C: 0%) vs. CD (F: 10%, P 10% & C: 80%) for 3 w	Improved short-term memory (Y-maze) but no changes in long-term memory (MWM)	↑ GluA1 levels in HPC (no changes in GluA2)	-	[182]
Adult animals	Mouse (♂; 3 m)	KD (F: 90%, C: 0% & P 10%) vs. CD (F: 10%, C: 80% & P 10%) for 3 m	No changes in learning (MWM) and memory (MWM and Y-maze)	No changes in LTP	-	[183]
	Rat (86% ♂; 4 and 20 m)	KD (F: 75.9%, C: 3.9% & P 20.1%) vs. CD (F: 16.4%, C: 64.9% & P 18.8%) for 12 w	Improved memory (WM/BAT)	-	↓ GLUT1 in PFC	[184]
	Rat (♂; 4 and 20 m)	KD (F: 75.9%, C: 3.9% & P 20.1%) vs. CD (F: 16.4%, C: 64.9% & P 18.8%) for 12 w	-	↓ expression of gria1, 2 and 4, and other postsynaptic proteins in DG-HPC (no changes in CA3)	↓ presynaptic proteins in DG-HPC (no changes in CA3)	[185]
	Mouse (♂; 12 m)	Cyclic KD: KD (F: 90% & P: 10%) & CD (F: 13%, P: 10% & C: 77%) alternate weekly; vs. CD (F: 14%, P: 24% & C: 62%) for 12 m	Improved memory (place avoidance test and NOR)	-	-	[186]
Diseased animals	Mouse (5 m; AD model and WT)	KD (F: 77.1%, C: 0.5% & P: 22.4%) vs. CD (F: 14%, C: 62.2% & P: 23.8%) for 3 m	No changes in memory (RAWM) but improved motor function (AD or WT animals)	No changes in neural loss	-	[187]
	Mouse (7 m; AD model and WT)	KD (F: 76%, C: 3% & P: 16%) vs. CD (F: 12%, C: 65% & P: 23%) for 2 & 4 m	Improved learning and memory at 4 m (no changes at 2 m; T and Barnes maze tests)	-	↓ Aβ, microgliosis, inflammation and ↑ number of spines and neurons in HPC	[188]
Standard diet supplemented with ketone bodies						
Humans	Human (♂ and ♀; ~15 y; T1D)	KB solution (40 g of MCT) vs. placebo drink in a single session of 1 h under hypoglycemic conditions (tests performed 1 h before & 1 h latter)	Improved hypoglycemia-mediated memory deficits	-	-	[189]
	Human (♂ and ♀; 35–70 y; T2D)	BHB (0.9% weight/volume) i.v., infusion vs. placebo (tests performed 120 min latter)	Improved working memory (no changes in global cognition)	-	-	[190]
	Human (♂ and ♀; 60–74 y)	Ketogenic formula (20 g of MCT in 36 g of total F) or placebo 90 min before tested	Improved working memory	-	-	[191]
	Human (♂ and ♀; ≥55 y; MCI)	Ketogenic solution (MCT) daily or placebo 6 m	Improved executive function, memory, and language	-	-	[192]
	Human (♂ and ♀; ~73 y; mild-to moderate AD)	Ketogenic formula (20 g of MCT in 35.9 g of total F) daily or placebo for 12 w vs. baseline	Improved in immediate and delayed logical memory	-	-	[193]
	Human (~74.7 y; mild-to moderate AD)	Ketogenic solution (MCT) vs. placebo solution (long chain triglycerides; tests performed 120 min latter)	Improved cognitive performance only in ApoEε4—subjects	-	-	[194]
Young animals	Mouse (♂ and ♀; 8 w)	BHB supplementation (60 mg/kg) for 2 days (twice/day) via intragastric gavage	-	↑ BDNF levels in HPC		[195]
Aged animals	Rat (♂; 21 m)	MCT8 formula (5 % of octanoic triglyceride) or MCT10 formula (5 % of decanoic triglyceride) vs. CD (5 % sunflower oil) for 8 w	Improved memory (NOR) with MCT10 (no changes with MCT8)	No changes in PSD95 and synaptophysin in brain. ↓ expression of plasticity-related genes	↑ INS signaling in brain	[196]
Diseased animals	Mouse (♂ and ♀; 8 w; AD model)	BHB solution (0.019 g/mL) daily for 8 w	-	-	↓ Aβ and inflammation in brain	[197]
	Mouse (4 m; AD model and WT)	BHB & ACA solution via subcutaneous injection (600 mg/kg/day & 150 mg/kg/day) daily for 2 m	Improved learning and memory (MWM and NOR; no changes in WT)	↑ LTP (no changes in WT)	↓ Aβ and oxidative stress in brain	[198]
	Mouse (8.5 m; AD model)	KE + BHB supplementation (ketone esters: 21.5%; F: 8.2% C: 43.5% & P: 23.9%) vs. CD (F: 8.2% C: 64.9% & P: 23.9%) for 4 & 7 m	Improved memory (MWM and FC)	-	↓ Aβ and phosphorylated tau in CA1 and CA3-HPC, amygdala and CTX	[199]
	Mouse (♂; 7.5 m; AD model)	HBME (10, 40 & 80 mg/kg/d) daily intragastric administration vs. water for 2.5 m	Improved learning and memory (MWM)	-	↓ Aβ in HPC and CTX. ↓ ROS and apoptosis under glucose deprivation in cultured neurons	[200]

KD and KB supplementation effects were classified according to experiment approach, taking into account species (human, rodent), sex, rodent age classified as young (≤2 months), adult (>2 months and ≤20 months), or aged (>20 months), and diseased animals (Alzheimer disease (AD) models). Search terms used were as follows: “ketogenic diet”, “ketone body” OR “medium-chain triglyceride” AND “learning”, “memory”, “synaptic plasticity” OR “AMPAR”. Only studies with humans and rodents were included, and only mild cognitive impairment (MCI) and AD pathological conditions were included. Excluded were human case reports and experimental models of maternal exposure. Percentage fats, carbohydrates, and proteins are relative to total energy unless otherwise indicated. Table abbreviations (used for the first time in this table) are as follows: ACA: acetoacetate; BHB: β-hydroxybutyrate; g: grams; *gria*: gene-coding glutamate ionotropic receptor AMPA type subunit; HBME: 3-hydroxybutyrate methyl ester; KE: ketone esters; MCTs: medium-chain triglycerides; ROS: reactive oxygen species; WT: wild type; WM/BAT: working memory/bi-conditional association task; and y: years.

## Data Availability

Not applicable.

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
