# Peer review of "Feeding the Brain: Effect of Nutrients on Cognition, Synaptic Function, and AMPA Receptors"

_nutrients, 2022, doi:10.3390/nu14194137_

Round 1

Reviewer 1 Report

In the present manuscript, the Authors summarize the extensive evidence linking nutrient excess and hippocampal plasticity, finally affecting several brain functions.

The authors focus on the effect of WD, KA, and PD on AMPAR, which are critical for neuroplasticity events. 

This review is a well-written summary focusing on relevant experimental work and recent advances in the field of nutrient effect on cognition.

Specific points to be addressed:

Page 2, line 57: Please, spell out the term "MetS."

Page 2, line 83: explain better the link between AD and AMPAR. Indeed, in AD the AMPARs are downregulated. 

Figure 3 and in the main text. The Authors should also consider the role of exosomes in hippocampal plasticity (e.g., A Vinuesa et al. Mol Neurobiol.2019).

Please try to conclude each chapter with some conclusions. Please try to link all the endings.

An attempt to be more speculative, with some research from a clinical and preclinical point of view where therapeutic strategies can counter or restore cognitive impairment in the metabolic syndrome

Reviewer 2 Report

Interesting narrative review that covers the topic in detail. The images are very enlightening. Well written. Although more tests should be used in humans, it is very important that even younger ages are affected and that ω-3 fatty acids can reverse most of the molecular deleterious effects of saturated fatty acids in HPC synapsis

Reviewer 3 Report

Accepted with minor revision 

the authors could add a section about the role of ketogenic diet in treatment of Alzheimer's disease explaining the role of ketone bodies in reversing the neuronal Insulin resistance 
